# The Comparison with Commercial Antioxidants, Effects on Colour, and Sensory Properties of Green Tea Powder in Butter

**DOI:** 10.3390/antiox12081522

**Published:** 2023-07-29

**Authors:** Songül Çakmakçı, İlhami Gülçin, Engin Gündoğdu, Hatice Ertem Öztekin, Parham Taslimi

**Affiliations:** 1Department of Food Engineering, Faculty of Agriculture, Atatürk University, Erzurum 25240, Türkiye; 2Department of Chemistry, Faculty of Science, Atatürk University, Erzurum 25240, Türkiye; igulcin@atauni.edu.tr; 3Department of Food Engineering, Faculty of Engineering and Natural Sciences, Gümüşhane University, Gümüşhane 29100, Türkiye; engingundogdu@gumushane.edu.tr; 4Department of Dairy Process Technology, Diyarbakır Agriculture Vocational School, Dicle University, Diyarbakır 21280, Türkiye; 5Department of Biotechnology, Faculty of Science, Bartın University, Bartın 74100, Türkiye; ptaslimi@bartin.edu.tr

**Keywords:** butter, oxidation, green tea powder, antioxidant, shelf life, butter analysis

## Abstract

Oxidation is one of the most important factors limiting shelf life and is a major deterioration process affecting both the sensory and nutritional quality of food. The high oxidation stability of lipids, which can be improved by the addition of antioxidants, is important for health protection, food quality, and economic reasons. In recent years, research on plant-derived antioxidants for use in human health and food has steadily increased. The aim of this study was to compare the antioxidant effects of green tea powder (GTP) in butter with those of commercial antioxidants (BHA, BHT, α-tocopherol, and Trolox). In addition, the effects on colour, sensory, gross physicochemical properties, and β-carotene content were investigated in butter. After the separation of butter into five pieces, the first part was chosen as the control sample without GTP; the second part has 100 mg/kg of BHT added to it; and the third, fourth, and fifth parts had 1, 2, and 3% of GTP added in the samples. They were stored at 4 ± 1 °C. Analysis was performed at intervals of 15 days. According to the iron reduction, CUPRAC and FRAP methods were performed, and parallel results were observed. Using the radical elimination methods (ABTS, DPPH^•^, and DMPD^•+^), IC_50_ values were calculated for the samples. According to the IC_50_ values, the GTP-containing samples were good antioxidants. The total phenolic andf β-carotene contents increased as the GTP addition increased. The addition of GTP had an antioxidant capacity equal to or higher than that of the BHT-added sample. For the production of a sensory-pleasing, greenish-coloured, new functional butter, the 1% GTP addition showed the most positive results.

## 1. Introduction

Butter is an essential dairy product in human nutrition because it has a pleasant aroma, melts at body temperature, is easily digested, contains vitamin A and/or β-carotene and essential fatty acids, and is a source of energy. It is a fat that can immediately lose its sensory and nutritional quality depending on time and conditions during storage. Oxidation is the most important factor limiting the shelf life of butter. The oxidation of butter causes a rancid taste and unpleasant odour, significant reductions in quality, and rejection by the consumer [1,2]. Toxic reaction products form in the latter stages of lipid oxidation. These products affect human health by causing various diseases [1,3]. Natural or synthetic antioxidants are used to prevent lipid oxidation. Consumers have turned to natural products due to the research results in which the adverse effects of synthetic additives/antioxidants on health were determined [4]. In recent years, research on plant-derived antioxidants has increased in terms of human health and use in foods [1,5,6,7]. Research is continuing to use natural antioxidants to prevent oxidation in butter. Ziarno et al. [8] determined that rosemary and thyme extracts had a significant beneficial effect on the oxidative stability of butter. The effects of methanol extracts of sage, thyme, and rosemary were investigated by Ayar et al. [9]; the effect of oregano essential oil was investigated by Dagdemir et al. [10]; the effect of black seed oil was investigated by Çakmakçı et al. [11]; and the effect of *Satureja cilicica* essential oil was investigated by Özkan et al. [12]. However, since these materials mask the aroma of butter due to their intense odours, their usability without solving the odour problem is poor. Therefore, this study aimed to study the effects of green tea powder (GTP), which has less odour and high antioxidative effects. Green tea (*Camellia sinensis* L.) is obtained from fresh tea leaves [13,14]. Green tea is the least processed tea product. It has a high antioxidant content due to polyphenols, especially catechins [15]. Green tea is an excellent source of water-soluble polyphenol antioxidants. The most important green tea catechin is epigallocatechin-3-gallate (EGCG), which is responsible for a wide range of health benefits [14,15,16,17,18]. EGCG is the most abundant and researched compound in tea [13,19]. Green tea reduces cancer risk [17], increases fat burning, and improves physical performance, and it may also protect the brain in old age, lowering the risk of Alzheimer’s and Parkinson’s diseases [13,20]. It can kill bacteria, improve dental health, and lower the risk of infection. Additionally, it may lower the risk of diabetes and cardiovascular disease. It can help in weight loss [17]. It shows characteristics such as antioxidant, antihypertensive, anticancer, antidiabetic, antibacterial, antiatherosclerotic, anti-inflammatory, etc. [17,20]. From a technological point of view, green tea and its products are inexpensive, natural, food-grade additives that inhibit the growth of foodborne pathogens and lipid oxidation, therefore increasing the shelf life of foods [21].

Given the proven antioxidant properties of green tea, we hypothesised that a GTP addition as an antioxidant ingredient in butter could prevent lipid oxidation in butter. To the best of our knowledge, this is the first study to show the impact of GTP as a potential antioxidant compared to butylated hydroxytoluene (BHT), a strong synthetic antioxidant, on the oxidation process of butter. In addition, the colour, sensory properties, and other butter properties were studied in this study. Thus, this research aims to combine the benefits of both butter and green tea into a new kind of butter. GTP may be an alternative to BHT as a natural herbal antioxidant source. GTP is thought to contribute to the production of a new greenish-coloured functional butter type, and the produced butter can be consumed more in breakfast.

## 2. Materials and Methods

### 2.1. Materials

Cow milk creams used in butter production were obtained from Acar Dairy Factory (Erzurum, Turkey). The *Lactococcus lactis* subsp. *cremoris*, *Lactococcus lactis* subsp. *lactis*, *Lactococcus lactis* subsp. *lactis* biovar *diacetylactis*, and *Leuconostoc mesenteroides* subsp. *cremoris* were used as the mixed starter culture. The code of this culture is DVS (50) CH Normal 22 (LD) and was obtained from Peyma-Hansen Cheese Rennet Industry and Trade Inc. (Istanbul, Turkey). Fresh tea leaves were harvested and processed at the Atatürk Tea and Horticultural Research Institute (Rize, Turkey). The harvested green tea leaves were steamed, rolled, dried, ground, and sieved. GTP forms smaller than 50 μm were obtained using a special sieve (Figure 1B). In the packaging of produced butter, three 250 g butter samples were first wrapped with stretch film and then packed with aluminium foil and stored separately for each storage period at 4 ± 1 °C. In addition, a spare sample of each sample was stored at −20 ± 1 °C. The colours of the produced butter are shown in Figure 1C (there are five samples in one replication: control, BHT-added butter, 1% GTP, 2% GTP, and 3% GTP-added butter samples).

### 2.2. Experimental Design

The definition of the butter samples produced in the research is given in Table 1. The research was carried out with two replications based on the factorial experimental design. The analyses were conducted in parallel. A total of 90 butter samples were produced; antioxidant properties, pH, colour, and sensory analysis were performed during nine storage periods (days 1, 15, 30, 45, 60, 75, 90, 105, and 120). Dry matter, fat, ash, and protein analyses were performed after production (first day).

### 2.3. Production of Butter Samples

The production flow chart of butter samples is shown in Figure 2.

### 2.4. Determination of Physicochemical Properties

Dry matter was analysed via oven drying at 102 ± 1 °C until constant weight and fat content was determined using the Gerber method, protein content was determined using the micro Kjeldahl method, and ash content was determined according to the method described by Kurt et al. [22]. The pH analyses were performed according to Atamer [23]. The pH values of the serum phase of butter were determined using a digital pH meter (Mettler-Toledo AG 8603 Schwerzenbach, Switzerland). 

### 2.5. Determination of the β-Carotene Amount

The extraction process applied to determine the amount of β-carotene in butter samples was performed according to the described method by Hulshof et al. [24] with slight modifications. The extraction method applied to milk was applied to butter (1 g) [24]. The β-carotene contents of the butter samples were analysed using the reversed-phase liquid chromatographic method described by Cakmakci et al. [25]. A Shimadzu LC-20AD Prominence HPLC system (Shimadzu Corp., Kyoto, Japan) was used, and the system consisted of a SPD-M20A diode array detector, a SIL-20A HT autosampler, a CTO-20A column oven, and a DGU-20A5 degasser. The wavelength of the detector was set at 450 nm. A Phenomenex Jupiter C18 column 250 × 4.6 mm × 5 mm (Phenomenex Co, Torrance, CA, USA) was used for separation. The solvent system, methanol-BHT stabilized THF 98/2 (*v*/*v*), was used as the mobile phase, and the flow rate was 1.0 mL/minute at an isocratic flow. The concentration of β-carotene in butter samples was calculated via the external standard method using a β-carotene standard (Sigma Aldrich, CAS: 7235-40-7) and expressed as micrograms of β-carotene per 100 g of the sample on a dry weight basis.

### 2.6. Determination of Antioxidant Activity

Chemicals such as ABTS, DMPD, DPPH, and standard compounds (α-tocopherol, BHA, BHT, and Trolox) were purchased from Sigma–Aldrich (Darmstadt, Germany) for use in antioxidant methods. Different concentrations (10–30 mg/mL) of the extracts and reference standards were used to examine the effect of the dose-dependent antioxidant potential of the extracts.

#### 2.6.1. Fe^3+^–Fe^2+^ Reduction Capacity

The reduction ability determination in the butter samples was made according to the slightly modified version of the Oyaizu method [26]. Before starting the experiments, stock solutions of the samples at 1 mg/mL concentrations were first prepared. Then, these solutions were diluted depending on the situation and used. For this aim, first, a 1 mg/mL stock solution was prepared. Different concentrations of this stock solution were transferred to glass tubes, and the solution was completed with 1 mL of distilled water. Then, 2.5 mL of phosphate buffer (0.2 M, pH 6.6) and 2.5 mL of 1% potassium ferricyanide [K_3_Fe(CN)_6_] were added, and this mixture was incubated at 50 °C for 20 min. After these procedures, 2.5 mL of 10% trichloroacetic acid (TCA) was added to the reaction mixture. Then, 2.5 mL of the supernatant of the solution was taken, after which 2.5 mL of distilled water and 0.5 mL of iron III chloride (FeCl_3_) (0.1%) were added. The absorbance was read against the blank sample at 700 nm. For preparing the control, distilled water was used instead of the sample.

#### 2.6.2. Cu^2+^–Cu^+^ Reduction Capacity (CUPRAC Method)

Cu^2+^ reduction activities in the butter samples were performed with a slight modification of the cupric ions’ reduction method [27,28]. A Copper (II) chloride (CuCl_2_) solution (0.25 mL, 0.01 M), 0.25 mL of ethanolic neocuproine solution (7.5 × 10^−3^ M), and 0.25 mL of ammonium acetate (CH_3_COONH_4_) and buffer solution (1 M) were added to the tubes of butter samples prepared at different concentrations. After a 30 min incubation, the absorbance values were measured at 450 nm against the distilled water.

#### 2.6.3. FRAP Reduction Activity

The stock solution prepared for the previous methods was also used in this experiment. First, the butter samples and standard solutions were transferred into the test tubes at 10, 20, and 30 µg/mL concentrations. The volumes were brought to 0.5 mL using a buffer solution. Then, a 2250 μL, 20 mM FeCl_3_ solution and a 2250 μL FRAP reagent were added to the test tubes, and the total volume was completed to 5 mL. The test tubes were mixed in a vortex, and after 10 min, their absorbance was recorded at 593 nm. An acetate buffer was used as a blind sample. The solution was left at room temperature for 10 min after vigorous mixing in the vortex. After incubation, the absorbance of the solution was recorded at 562 nm against ethanol. The remaining solution was used as a control, excluding the sample with phenolic compounds [29].

#### 2.6.4. 1,1-Diphenyl 2-picrylhydrazyl (DPPH) Free Radical Scavenging Activity

The DPPH free radical scavenging activity of the butter samples was determined according to the Blois method [30]. A 10^−3^ M solution of DPPH was used as a free radical. The stock solution of the butter samples at a concentration of 1 mg/mL, which was prepared before, was used. Solutions were transferred into the test tubes at concentrations of 10, 20, and 30 µg/mL, and 1 mL of stock DPPH solution was added into each sample tube after completion with ethanol, with a total volume of 3 mL. After incubating for half an hour at room temperature and in the dark, the absorbance at 517 nm was measured against ethanol. As a control, 3 mL of ethanol and 1 mL of the DPPH solution were used. The reduced absorbance gave the remaining DPPH solution, i.e., free radical scavenging activity.

#### 2.6.5. 2,2′-Azino-bis(3-ethylbenzo-thiazoline-6-sulfonic acid) (ABTS) Radical Scavenging Activity

The ABTS radical scavenging activity of the butter samples was determined according to Re et al. [31]. First, a 7 mM ABTS solution was prepared. ABTS radicals were produced by adding 2.45 nM of persulfate solution to this solution. Before the ABTS radical solution was used, the absorbance of the control solution at 734 nm was adjusted to 0.700 ± 0.025 nm using a phosphate buffer (pH 7.4 and 0.1 M). One millilitre of ABTS radical solution was added to different concentrations (10–30 µg/mL) of the butter samples and incubated for half an hour. Absorbance was recorded at 734 nm against ethanol.

#### 2.6.6. N,N′-Dimethyl-p-phenylenediamine Dihydrochloride (DMPD) Radical Scavenging Activity

This DMPD radical scavenging activity was determined according to the method given by Fogliano et al. [32]. For this purpose, the coloured radical cation (DMPD^•+^) was first obtained. For this, 100 mL of DMPD solution (pH 5.3; 100 mM) was obtained by adding 0.2 mL of FeCl_3_ to become 0.05 M. Measurements were made at 505 nm for 1 mL of this solution. Before using the DMPD radical solution, the optical density of the control solution at 505 nm was adjusted to 0.900 ± 0.100 nm using a phosphate buffer (0.1 M and pH 5.3). The absorbance of the daily, freshly prepared DMPD^•+^ solution was constant for up to 12 h. The solutions of the butter samples and standard antioxidants at different concentrations (10–30 µg/mL) were transferred into the test tubes, and the volume was brought to 0.5 mL using distilled water. Then, 1 mL of DMPD^•+^ solution was added to it. After a 50 min incubation, the absorbance values were measured at 505 nm against the buffer solution.

### 2.7. Determination of Phenolic Substances

The total phenolic content (TPC) in the samples was determined by measuring the colour of the phenolic compounds with a spectrophotometer via a Folin–Ciocalteu solution in an alkaline medium [33]. For this, the appropriately diluted sample was taken, and the Folin–Ciocalteu solution (1 mL) was left for 5 min, and then sodium carbonate (Na_2_CO_3_) solution (0.5 mL, 1%) was added and then incubated for 30 min. The absorbance of the mixtures was measured at 760 nm after incubation for 30 min at room temperature. The calculation was made using the standard curve drawn with gallic acid, and the TPC amounts were given as μg GAE/mg of dry weight.

### 2.8. Colour Measurement

The colour values (*L**, *a**, *b** values) of the butter samples were measured via a Minolta colorimeter (Chroma Meter, CR-200, Osaka, Japan) and a Konica Minolta [34]. Measurements in the samples were calculated by averaging the colour values read from three to five different points. Before the measurements were made, the instrument was calibrated with *a** white calibration plate, and all measurements were performed on a white background using chrome drying containers.

### 2.9. Sensory Evaluation Analysis

Sensory evaluation (colour, odour, texture, flavour, bitter/rancid taste, and general acceptability) was performed with thirty consumers (students and teaching staff of the Atatürk University Food Engineering Department, and other consumers: age: 18–50 years) using a 9-point hedonic scale (1 = dislike extremely, 9 = like extremely). The panellists were experienced and familiar with the taste of butter and green tea. In addition, all the participants were trained and assured that they understood the definitions of colour, texture, odour, flavour, and bitter/rancid taste before the panel test for the butter. At 4 °C, the different kinds of butter samples, approximately 30 g, were stored at 15–18 °C, wrapped in aluminium foil and coded with three-digit numbers. The tests were conducted under fluorescent light. Warm water and plain bread were used to rinse their palates between the butter samples. The panellists were not allowed to discuss with each other during the sensory evaluation and were asked to complete the form within 30 min.

### 2.10. Statistical Analysis

Research trial design: 5 different applications (Table 1) × 9 storage periods (1, 15, 30, 45, 60, 75, 90, 105, and 120 days) × 2 (repetition) were carried out according to the factorial experimental design. Thus, the number of samples was 90. The analyses were conducted in parallel. The obtained data were subjected to variance analysis, and Duncan’s multiple comparison tests were applied to the averages, which were found to be significant. Statistical analysis was performed using the SPSS software package (Version 9.0, Chicago, IL, USA).

## 3. Results and Discussion

### 3.1. Properties of Materials Used in Butter Production

The analysis results of cow milk cream and GTP used in butter samples are given in Table 2. It was determined that the dry matter (92.70%), protein (19.44%), and ash (4.13%) amounts of GTP were very high. The cream’s fat, protein, and ash ratios are 50%, 0.80%, and 0.60%, respectively. Therefore, as the amount of GTP added in the butter increased, the amount of these components in the butter samples also increased (Table 2).

### 3.2. Physicochemical Properties of the Butter Samples

The analysis results of the dry matter, fat, protein, and ash made on the first day after the production of the butter samples are given in Table 2. The dry matter, fat, and protein ratios of the control group and BHT-supplemented butter samples showed no statistically significant differences. As the ratio of the GTP added to butter increased, the protein and ash ratios of butter increased. These increases (excluding 1% GTP in the protein and ash amounts) were statistically significant and different from each other. This result is because the ratio of protein (19.44%) and ash (4.13%) of the GTP was higher than that of the control butter (Table 2). Chacko et al. [19] also stated that green tea has high protein (15–20% dry weight) and mineral (5%) contents. A significant result is that 1% GTP does not increase the protein content. This result is positive and essential since the protein spoils the quality of butter. This result is essential in the overall assessment in terms of the recommendation of the GTP1 sample. As the fat content (7%) of the GTP was lower than that of butter [19], the amount of fat was lower due to the GTP being added to the butter. The pH values found in the butter samples during storage are shown in Figure 3. As the GTP ratio increased, the pH increased compared to the control samples (C and with BHT), and in the following periods, the highest pH values were found in the 2% GTP samples and decreased in all samples. It was determined that pH values decreased in all samples during storage (Figure 3). The average β-carotene contents are given in Table 2. The β-carotene amounts of the C and BHT samples were not significantly different from each other. However, as the amount of GTP increased, the amount of β-carotene in butter also increased, and this increase was statistically significant.

### 3.3. Colour Values of Butter Samples

*L**, *a**, and *b** values are obtained by measuring the colour in three dimensions and giving the specific colour value. The *L**, *a**, and *b** values denote lightness–darkness, redness–greenness, and yellowness–blueness, respectively. Higher *L** values indicate lighter butter colour, whereas lower *L** values indicate darker butter colour. A negative *a** value indicates greenness, whereas a positive *a** value indicates redness. A lower negative b* value indicates more blueness, whereas a higher positive *b** value indicates more yellowness [16]. The butter samples were produced from cow milk cream. In a previous study, the amount of β-carotene was higher in the cow milk-derived butter, and it was not found in sheep and goat milk butter [35]. Therefore, the colour of cow butter is yellowish. This study took different shades of the greenish colour, except for the control and BHT samples, as different levels of GTP were used, and its colour was distinctly green. The colour darkened due to the increased concentration of the GTP. According to this, the brightest butter samples were C and BHT samples. The *L**, *a**, and *b** values in the C and BHT samples showed no statistically significant differences, and this result was typical because BHT is a white substance. In the GTP samples, *L** and *a** values were lower, *b** values were higher depending on concentration, and the colour values were different (C and BHT samples were not different) (Table 3). Slight fluctuations in the colour values were observed during storage. During storage, the samples were found to be lower than the values at the beginning of storage.

### 3.4. Antioxidant Analysis Results

In this section, studies on determining the antioxidant capacities of the butter samples are given. Antioxidant activity works with different mechanisms, such as radical removal, inhibition of lipid peroxidation, metal chelation, and a reduction in oxidising agents. Many different methods have been designed to determine the reduction capacity. In this research, the reduction capacity in ferric ions (Fe^3+^) to ferrous ions (Fe^2+^), the reduction capacity in cupric ions (Cu^2+^) to cuprous ions (Cu^+^), and the reduction capacity in Fe^3+^-TPTZ via the FRAP method were studied. These methods determined the antioxidant capacities of butter containing GTP at different concentrations. The capacity of a compound to reduce ferric ions (Fe^3+^) is an essential indicator of its potential antioxidant activity [36]. According to the reduction methods, iron reduction, CUPRAC, and FRAP methods were performed, and similar results were observed in all three methods. Comparisons have been made with synthetic and standard antioxidants such as BHA, BHT, α-tocopherol, and Trolox, an analogue of α-tocopherol. After the 60th day of storage, the antioxidant capacities decreased, but almost the same and better results were found compared to butter containing BHT (Figure 4A). The results were good between 1 and 90 days, and then there were some changes. Antioxidant results were as we wanted in these three parameters including ferric ions (Fe^3+^) to ferrous ions (Fe^2+^), cupric ions (Cu^2+^) to cuprous ions (Cu^+^), and FRAP). The CUPRAC method was used to determine the reduction force in the butter samples with GTP added, which showed a similar reduction strength according to the standards. According to the FRAP method, the capacity of butter to reduce ferric ions (Fe^3+^) to ferrous ions (Fe^2+^) increased in direct proportion to the concentration (Figure 4A). Butter containing GTP showed good antioxidant properties. In all three reduction methods, it was determined that the reduction capacity in butter increased depending on the GTP concentration (Figure 4A). The most popular spectroscopic methods used to determine the radical scavenging activity of antioxidants are DPPH, ABTS^•+^, and DMPD+ radicals. In particular, the ABTS^•+^ and DPPH methods are simple, fast, sensitive, and reproducible [5]. For these internal parameters (DPPH, ABTS, and DMPD), IC_50_ values between 1 and 75 days started to decrease, which is valuable for us. The IC_50_ values started to increase, and the antioxidant results slightly decreased. In most antioxidant methods, GTP activity has similar activities to the activities of the standards used. This study shows the IC_50_ values related to the DPPH radical scavenging activity of GTP in butter in Figure 4B. The IC_50_ value of the DPPH radical scavenging activity is 34.9 µg/mL for curcumin [28], 6.96 µg mL^−1^ for resveratrol [37], and 31.81 µg/mL for propolis [38]. The results of the mentioned studies, when compared with the GTP in this study, the low IC_50_ values show that GTP is also an excellent antioxidant in butter (Figure 4B). ABTS^•+^ scavenging assay is a beneficial method that is soluble in water and organic solvents, as it is not affected by ionic strength. In this method, similar results were found. Antioxidant activity works with different mechanisms, such as radical scavenging, inhibiting lipid peroxidation, metal chelation, and reducing oxidizing agents. Therefore, it is not surprising that in the butter samples with the different GTP rates added, the reduction forces are high depending on the percentage. However, while the TBA results are expected to decrease depending on this, the reason for the relatively higher result is estimated to be due to an extra 7% lipid composition found in the dried tea samples [19].

According to the IC_50_ values, butter containing GTP is a good antioxidant and radical scavenger. In these methods, the results were similar according to BHT, and the IC_50_ values were found to be small in butter with GTP, so it was found that they were good antioxidants. A slight antioxidant capacity decrease or an IC_50_ increase in these methods was observed after 60 days (Figure 4B). The antioxidant effect order of the GTP amounts added to the butter was found as follows: 1% < 2% < 3%. As a result of the research, very successful results were obtained. GTP can be used as a natural antioxidant that is even more effective than BHT in butter.

### 3.5. Total Phenolic Content

Phenolics are the most active natural antioxidants, and their antioxidant effects are achieved by binding free radicals, forming chelates with metals, and inhibiting lipoxygenase enzymes. The TPCs for the control butter sample, 100 mg/kg of BHT, and butter samples containing 1, 2, and 3% of GTP are given in Figure 5. Accordingly, as the GTP concentration increased, the TPC of the butter samples increased significantly (*p* < 0.01). The lowest TPC was determined in the C sample (48.52 µg GAE/mg), while the highest TPC was detected in the GTP3 sample (157.63 µg GAE/mg) (Figure 5). All samples containing GTP had a higher TPC than butter containing BHT. These results again confirmed that GTP could be used as an alternative to BHT as a natural source of antioxidants. According to the results obtained in this study, there is a positive correlation between TPC and antioxidant activity.

### 3.6. Sensory Evaluation

Both the flavour and colour of butter are important factors affecting consumer preference. The butter variety was found to be statistically significant at the level of *p* < 0.01 for all sensory characteristics (Table 4). According to Duncan’s multiple comparison test results, all sensory properties examined in the C and BHT samples showed no statistically significant differences. In addition, these values were found to be higher than the values of all the samples with the GTP; in other words, the panellists appreciated them more. In terms of GTP-added butter samples, scores were lower in parallel to the increase in the GTP rate. These decreases (except bitterness) were found to be significantly different from each other (Table 4). The most similar values for C and BHT samples were the same for the 1% GTP-added butter sample; texture, flavour, bitterness, and general acceptability scores were found to be approximately 7, which was evaluated as good. Colour (6.96) and odour (6.68) were good scores. The highest colour score among the samples with GTP was obtained with the 1% GTP butter sample. 

Even on the 120th day of storage, it received over 7 points. According to the Duncan’s multiple comparison test results of the butter type, the highest colour scores were obtained in control, BHT, and GTP1 samples (Table 4). This result can be attributed to the consumer’s familiarity with white–yellowish butter. Similar results in the colour of butter have been reported at different doses of various vegetables that are different from butter and are dark in colour [11]. Texture scores were found to be higher in C, BHT, and GTP1 butter samples; the reason for this is that GTP gives textures a slightly rough structure. No significant differences were observed in the same samples during storage (above 7) but were lower in GTP2 and GTP3 samples. A mild foreign flavour (6.2) from the GTP is typical; in fact, the fact that flavour and general acceptability scores are above 7 confirms this (Table 4). In other words, it is thought that the cause of the bitter taste is the GTP itself, which is not caused by oxidation. It was found that GTP masked the bitterness of oxidation and highlighted its bitterness (bitter taste). Liu et al. [16] also stated that the primary flavours of green tea infusion are bitterness and astringency. It was found that the 3% GTP-added butter samples would not be suitable for consumption, and the 1% GTP level was the most appropriate level from the general evaluation of all properties. This result was consistent with some previous studies [10,11].

The C and BHT samples had the highest scores, while the GTP2 and GTP3 butter samples generally scored lower. The results closest to the controls (C and BHT samples) during storage are marked in the GTP1 sample. This had reduced the overall acceptability since the distinctive taste and odour from the GTP are different from the usual flavour of butter. The most effective factor in general acceptability is flavour. The GTP’s bitterness also influenced this result, and the addition of 1% GTP did not result in significant bitterness and texture changes. The higher GTP content dissolved more flavour ingredients in butter, leading to more intense bitterness and a reduced overall acceptance. In particular, the GTP3 sample received a low score (Table 4). The general acceptability scores of the butter samples that were determined during storage are given in Table 4. The C and BHT samples had the highest scores, while the GTP2 and GTP3 butter samples generally had low scores that were not acceptable (Table 4). This may be due to the distinct bitter taste arising from GTP, which is different from the usual flavour of butter. As the GTP ratio increases, the darkness and opacity of the colour, roughness, and tea bitterness are thought to be the factors affecting it. The closest result to the controls is marked in the GTP1 sample. During storage, the closest scores to the controls were found in the GTP1 sample (approximately 7/9 points). As a result of sensory analysis, it was found that sensory properties were adversely affected as storage time generally progressed. However, in terms of consumption, values were generally in the range of 6–7 (excluding textures and tea bitterness) until the 105th day.

## 4. Conclusions

In this research, successful results were obtained. As a result of the antioxidant analysis, the samples with GTP had antioxidant capacities equal to or higher than that of the BHT-added sample. The sequence of antioxidant effect is as follows: 1% < 2% < 3% GTP. There was a positive correlation between TPC and antioxidant activity. TPC increased with the addition of the GTP. When the butter analysis results are evaluated collectively, the 1% GTP additive can be recommended to produce a new functional butter with a greenish, naturally preserved, and pleasant flavour. This research made it possible to produce functional and coloured butter varieties and increase the shelf life with natural vegetable material. It has also been seen that consumers, especially those who know the benefits of green tea, will generally encounter the slightly bitter taste caused by the GTP. Thus, GTP is a natural antioxidant that can be successfully used instead of BHT in butter.

## Figures and Tables

**Figure 1 antioxidants-12-01522-f001:**
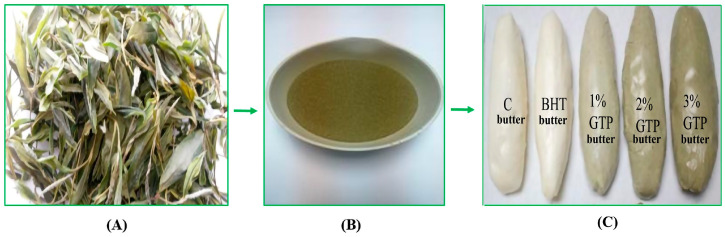
Dry green tea (**A**), green tea powder (GTP) (**B**), and fresh butter varieties (**C**).

**Figure 2 antioxidants-12-01522-f002:**
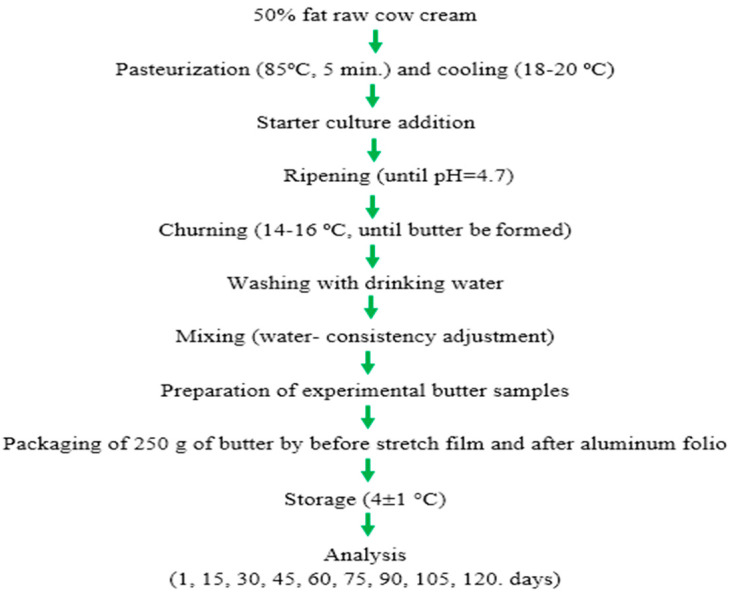
The production flow chart of butter samples.

**Figure 3 antioxidants-12-01522-f003:**
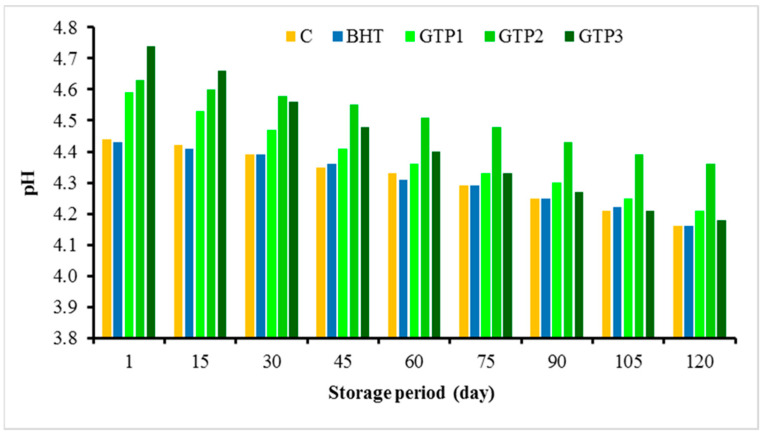
pH values of butter varieties during the storage period.

**Figure 4 antioxidants-12-01522-f004:**
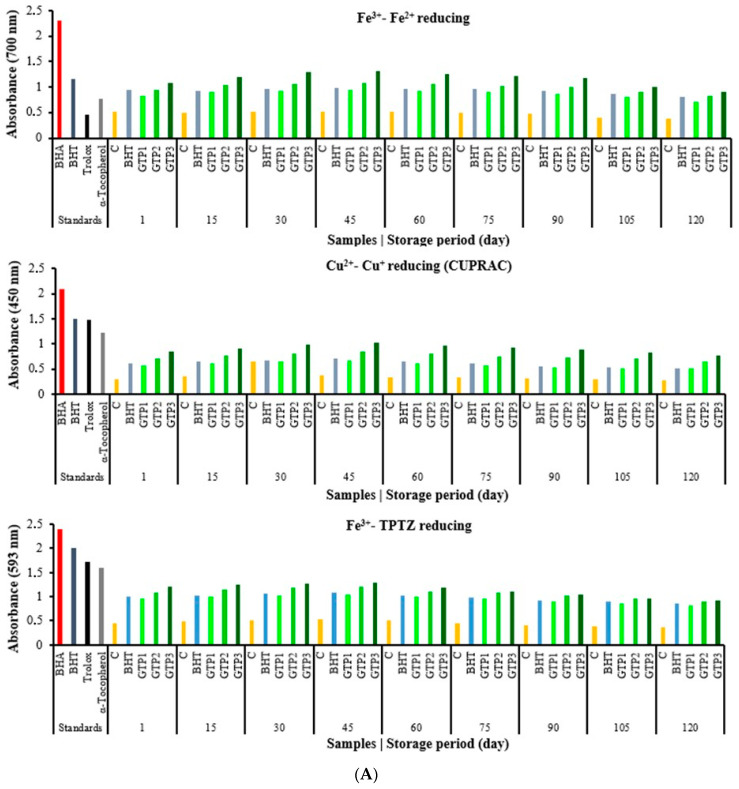
(**A**). Reducing ability results of butter varieties during the storage period. (**B**). Scavenging activity results of butter varieties during the storage period.

**Figure 5 antioxidants-12-01522-f005:**
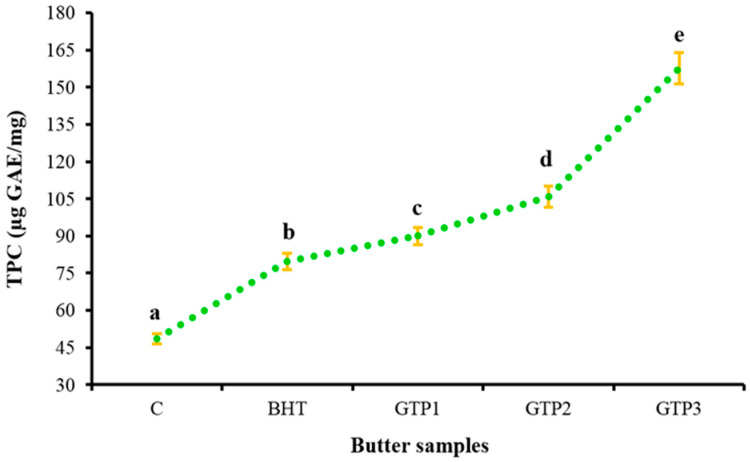
Total phenolic compounds (µg GAE/mg) in butter samples on the 1st day of storage. The averages indicated by different letters were significantly different from each other (*p* < 0.01).

**Table 1 antioxidants-12-01522-t001:** Formulation of experimental butter samples.

Sample Code	Sample Definition
C	Control butter [Control sample, no GTP added)
BHT	100 mg/kg BHT added butter (Control butter + 100 mg/kg BHT)
GTP1	1% GTP added butter (Control butter + 1% GTP)
GTP2	2% GTP added butter (Control butter + 2% GTP)
GTP3	3% GTP added butter (Control butter + 3% GTP)

**Table 2 antioxidants-12-01522-t002:** Characteristics of cream, GTP, and butter samples.

Properties	Raw Materials		Butter Samples *^,^**				
	Cream	GTP	C	BHT	GTP1	GTP2	GTP3
Dry matter (%)	-	92.70	83.50 ± 0.19 ^a^	83.59 ± 0.15 ^a^	82.41 ± 0.06 ^d^	82.61 ± 0.05 ^c^	82.81 ± 0.08 ^b^
Fat (%)	50.00	nd	82.13 ± 0.25 ^a^	82.25 ± 0.29 ^a^	81.25 ± 0.29 ^b^	80.38 ± 0.25 ^c^	80.00 ± 0.00 ^d^
Protein (%)	0.80	19.44	0.60 ± 0.01 ^c^	0.60 ± 0.01 ^c^	0.61 ± 0.04 ^c^	0.70 ± 0.01 ^b^	0.76 ± 0.02 ^a^
Ash (%)	0.60	4.13	0.53 ± 0.01 ^bc^	0.52 ± 0.01 ^c^	0.55 ± 0.01 ^bc^	0.57 ± 0.01 ^ab^	0.59 ± 0.01 ^a^
pH	4.33	5.20	4.44 ± 0.03 ^d^	4.43 ± 0.02 ^d^	4.59 ± 0.02 ^c^	4.63 ± 0.02 ^b^	4.74 ± 0.01 ^a^
β-carotene	-	-	1.57 ± 0.02 ^c^	1.56 ± 0.02 ^c^	1.96 ± 0.22 ^bc^	2.33 ± 0.38 ^ab^	2.53 ± 0.09 ^a^

* At the first day, average values of two replication; ** See Table 1 for descriptions of butter samples. The averages indicated by different letters within the same line are significantly different from each other (*p* < 0.01); nd: not determined.

**Table 3 antioxidants-12-01522-t003:** Results of colour values analysis of the butter samples.

		*L**	*a**	*b**
Butter type *	C	87.42 ± 2.30 ^a^	−3.45 ± 0.57 ^c^	12.71 ± 0.46 ^d^
	BHT	87.21 ± 2.42 ^a^	−3.35 ± 0.14 ^c^	12.77 ± 0.43 ^d^
	GTP1	72.40 ± 1.44 ^b^	−4.32 ± 0.34 ^b^	15.86 ± 0.59 ^c^
	GTP2	65.74 ± 1.24 ^c^	−4.43 ± 0.32 ^ab^	16.64 ± 1.79 ^b^
	GTP3	62.19 ± 1.81 ^d^	−4.47 ± 0.24 ^a^	17.40 ± 0.81 ^a^
Storage period (day)	1	76.01 ± 12.25 ^b^	−3.75 ± 0.45 ^d^	15.57 ± 2.17 ^a^
	15	74.42 ± 11.34 ^cd^	−3.79 ± 0.41 ^d^	15.19 ± 1.85 ^ab^
	30	76.83 ± 11.34 ^a^	−4.10 ± 0.76 ^ab^	14.48 ± 1.72 ^cd^
	45	75.33 ± 12.11 ^b^	−3.87 ± 0.42 ^cd^	14.83 ± 1.88 ^bc^
	60	75.47 ± 10.76 ^b^	−4.07 ± 0.53 ^ab^	14.28 ± 2.72 ^d^
	75	74.20 ± 10.02 ^d^	−4.15 ± 0.63 ^ab^	15.20 ± 2.23 ^ab^
	90	73.55 ± 9.87 ^d^	−3.99 ± 0.63 ^bc^	15.29 ± 2.38 ^ab^
	105	75.14 ^b^ ± 9.74 ^c^	−4.26 ± 0.76 ^a^	15.42 ± 2.40 ^a^
	120	73.97 ± 11.07 ^d^	−4.09 ± 0.67 ^ab^	15.45 ± 2.28 ^a^

* See Table 1 for butter sample descriptions. The average values followed by a different letter in a column (each section separately) were significantly different (*p* < 0.01).

**Table 4 antioxidants-12-01522-t004:** Results of sensory analysis of the butter samples.

		Colour	Texture	Odour	Flavour	Bitterness	Overall Acceptability
Butter type *	C	7.65 ± 0.52 ^a^	7.99 ± 0.49 ^a^	7.05 ± 0.62 ^a^	7.43 ± 0.9 ^a^	7.60 ± 0.95 ^a^	7.52 ± 0.64 ^a^
	BHT	7.65 ± 0.52 ^a^	8.16 ± 0.29 ^a^	6.92 ± 0.37 ^ab^	7.59 ± 0.67 ^a^	7.75 ± 0.63 ^a^	7.76 ± 0.41 ^a^
	GTP1	6.96 ± 0.39 ^b^	7.63 ± 0.41 ^b^	6.68 ± 0.34 ^b^	7.14 ± 0.4 ^b^	7.22 ± 0.83 ^b^	7.12 ± 0.5 ^b^
	GTP2	6.15 ± 0.92 ^c^	7.29 ± 0.45 ^c^	6.04 ± 0.76 ^c^	5.47 ± 0.88 ^c^	7.22 ± 0.76 ^b^	4.65 ± 1.25 ^c^
	GTP3	5.37 ± 1.04 ^d^	6.65 ± 0.64 ^d^	5.75 ± 0.83 ^c^	3.41 ± 0.33 ^d^	7.24 ± 0.41 ^b^	3.43 ± 0.81 ^d^
Storage period (day)	1	6.88 ± 1.4 ^a^	7.95 ± 0.76 ^a^	6.31 ± 0.99 ^bc^	6.13 ± 2.04 ^b^	7.59 ± 0.66 ^b^	5.76 ± 2.50 ^c^
	15	6.77 ± 1.43 ^a^	7.99 ± 0.59 ^a^	6.42 ± 1.17 ^abc^	6.46 ± 2.05 ^ab^	7.64 ± 0.97 ^b^	6.25 ± 2.14 ^ab^
	30	6.75 ± 1.12 ^a^	7.70 ± 0.59 ^ab^	6.54 ± 0.74 ^abc^	6.53 ± 1.94 ^a^	8.07 ± 0.45 ^a^	6.64 ± 1.65 ^a^
	45	6.78 ± 1.0 ^a^	7.60 ± 0.65 ^b^	6.66 ± 0.91 ^ab^	6.42 ± 1.99 ^ab^	7.38 ± 0.94 ^b^	6.31 ± 1.89 ^ab^
	60	6.79 ± 0.54 ^a^	7.23 ± 0.53 ^cd^	6.49 ± 0.26 ^abc^	6.39 ± 1.8 ^ab^	7.59 ± 0.28 ^b^	6.14 ± 1.88 ^b^
	75	7.13 ± 0.48 ^a^	7.51 ± 0.45 ^bc^	6.82 ± 0.25 ^a^	6.12 ± 1.52 ^b^	7.18 ± 0.38 ^b^	6.35 ± 1.22 ^ab^
	90	6.92 ± 0.56 ^a^	7.43 ± 0.48 ^bcd^	6.72 ± 0.2 ^ab^	6.24 ± 1.46 ^ab^	7.53 ± 0.35 ^b^	6.28 ± 1.35 ^ab^
	105	6.76 ± 1.3 ^a^	7.41 ± 0.78 ^bcd^	6.29 ± 0.84 ^bc^	6.24 ± 1.58 ^ab^	7.45 ± 0.49 ^b^	6.12 ± 1.99 ^b^
	120	6.10 ± 1.86 ^b^	7.08 ± 1.08 ^d^	6.14 ± 1.07 ^c^	5.35 ± 1.53 ^c^	6.19 ± 0.54 ^c^	4.98 ± 2.42 ^d^

* See Table 1 for butter sample descriptions. The averages indicated by different letters in a column (each section separately) were significantly different from each other (*p* < 0.01).

## Data Availability

The data presented in this study are available on request from the corresponding author.

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
