# Peer review of "The Comparison with Commercial Antioxidants, Effects on Colour, and Sensory Properties of Green Tea Powder in Butter"

_antioxidants, 2023, doi:10.3390/antiox12081522_

Round 1
Reviewer 1 Report
Comment and suggestions for authors to improve the manuscript:
In my opinion, the abstract must be revised. The main objective of the work is not clearly shown, and from lines 17 to 21, the authors only mentioned part of methods applied.
In Introduction, references from 14 to 18 must be included as [14-18] in Line 55. Green tea obtained from tea leaves can result in a bitter taste, especially after minutes of preparation. Did the authors consider this possibility? In some works reported, the proposal of introducing herbs in butter or other food, including tea, is not new, therefore, which is the main novelty of this work respect to others?
Materiasl and Methods: The paragraph The starter culture used was Lactococcus lactis subsp. cremoris, Lactococcus lactis subsp. lactis, Lactococcus lactis subsp. lactis biovar diacetylactis and Leuconostoc mesenteroides subsp. cremoris is a mesophilic aromatic starter culture cannot understand well. Can it be revised?
In Figure 1C, 10 samples of fresh butter are shown, actually, how many different varieties are? Are these the control samples?
Which are the modifications in the method described by Hulshof et al. to determine the β-carotene?
I think it is better the use of the word blank instead of bleach.
I suppose that the value of scale for dislike extremely is 1 not 10.
The number of experiments seem to be excessive. Was not possible to design the experiments with other type of design?
Resuts and discussion:
As I commented before, I have doubts with the control samples. Thus, the values of composition in Table 2 corresponded to butter after 1 day, but for how many samples are the average values?
In Table 2, authors must include the meaning for nd. In any case, authors mentioned a value of 7% for fat content in Line 239, then, was it determined or not? It is confusing.
Part of the results mentioned in section 3.1 are repeated in section 3.2. Authors must revise it and correct it.
Figure 2 in Line 241 must be Figure 3.
From the results, I cannot understand the reason for the higher pH for the butter with a 2% of GTP, and not for 3%. Neither, why did it decrease with the storage period? Can the authors explain it?
What is the explanation for this paragraph While the amount of β-carotene was higher in cow milk-derived butter, it was not found in sheep and goat milk butter. There is no relationship with the content of the article, because only butter derived of cow milk was used.
Authors must revise these two paragraphs for understanding the results: In general, an increase in L* and a* values and b* values did not change. During storage, the samples were found to be lower than the values at the beginning of storage. An increase means a change, and the samples cannot be lower than values.
In Table 3, for which butter are the values for colour analysis during the storage days?
Antioxidant capacities must be given in the corresponding units of concentration depending of the method, but not in absorbance. In this way, it is impossible to compare with other values published for other authors.
If score for the butter with 45 days of storage is not significant with respect to that for 30 days, why is better to select this last?
What do the authors think about the green colour of the butters? Although the score is not bad, could it be a limitation for consumption?
Some of these products with tea extracts are already in the market, which are the following step in this research?
23 of the total references are older than 2013. Are all important for the manuscript?
In my opinion, it is necessary a revision of language. Some paragraphs are difficult to understand.
Author Response
Reviewer 1 (Answers of Authors)
Comments and Suggestions for Authors
Comment and suggestions for authors to improve the manuscript:
Responses to the First Referee's comments are notated in red (in the revised manuscript and letter).
In my opinion, the abstract must be revised. The main objective of the work is not clearly shown, and from lines 17 to 21, the authors only mentioned part of methods applied.
Author: The abstract was improved in the revised manuscript.
In Introduction, references from 14 to 18 must be included as [14-18] in Line 55. Green tea obtained from tea leaves can result in a bitter taste, especially after minutes of preparation. Did the authors consider this possibility? In some works reported, the proposal of introducing herbs in butter or other food, including tea, is not new, therefore, which is the main novelty of this work respect to others?
Author: The referee is right. However, since the writing rules of the Journal of Antioxidants and sample articles were requested to be written separately, no changes were made. Since the references are shown as numbers, those in the references list are easily seen in the article text.
Materiasl and Methods: The paragraph The starter culture used was Lactococcus lactis subsp. cremoris, Lactococcus lactis subsp. lactis, Lactococcus lactis subsp. lactis biovar diacetylactis and Leuconostoc mesenteroides subsp. cremoris is a mesophilic aromatic starter culture cannot understand well. Can it be revised?
Author: The sentence was corrected in the revised article.
In Figure 1C, 10 samples of fresh butter are shown, actually, how many different varieties are? Are these the control samples?
Author: There are 5 samples in one replication: Control, BHT added butter, 1% GTP, 2% GTP and 3% GTP added butter samples, respectively. The other 5 samples are the 2nd replication samples.
This explanation has been added to the revised article
Which are the modifications in the method described by Hulshof et al. to determine the β-carotene?
Author: (Section 2.5 was added in the revised text)
The extraction method applied to milk was applied to butter (1 g) (Hulsof et al., 2006). The β-carotene contents of the butter samples were analyzed using a reversed-phase liquid chromatographic method described in Cakmakci et al. (2014) [25]. A ShimadzuLC-20AD Prominence HPLC system (Shimadzu Corp., Kyoto, Japan) was used and the system consisted of a SPD-M20A diode array detector, a SIL-20A HT auto sampler, aCTO-20A column oven and a DGU-20A5 degasser. The wavelength of detector was set at 450 nm. A PhenomenexJupiter C18 column 250 × 4.6 mm × 5 mm (PhenomenexCo, Torrance, CA) was used for separation. The solvent system, methanol-BHT stabilized THF 98/2 (v/v), was used as the mobile phase and the flow rate was 1.0 mL/minute at isocratic flow. The concentration of β-carotene in butter samples was calculated by the external standard method using a β-carotene standard (Sigma Aldrich, CAS:7235-40-7) and expressed as micrograms of β-carotene per 100 gram of sample on a dry weight basis.
I think it is better the use of the word blank instead of bleach.
Author: As the referee stated, we used the word "blank", not the word "bleach".
I suppose that the value of scale for dislike extremely is 1 not 10.
Author: Excuse me, it is my bad. Thank you, I changed in the revised text.
The number of experiments seem to be excessive. Was not possible to design the experiments with other type of design?
Author: There are 5 samples in one replication: Control, BHT added butter, 1% GTP, 2% GTP and 3% GTP added butter samples, respectively. The other 5 samples are the 2nd replication samples. Analyzes were performed at 15-day intervals to see the changes precisely. Therefore, the experiment was not too complicated and it was tried to increase the reliability of the results by working with many methods.
Resuts and discussion:
As I commented before, I have doubts with the control samples. Thus, the values of composition in Table 2 corresponded to butter after 1 day, but for how many samples are the average values?
Author: The explanations in Table 2 have been corrected in the Revise article. In Table 2, only the analysis results for the parameters performed on the first day are presented. These values are the average of 2 replications (in each replication, 2 parallel studies were studied, in short, they are the average of 4 numbers. see: 2.2. Experimental design).
In Table 2, authors must include the meaning for nd. In any case, authors mentioned a value of 7% for fat content in Line 239, then, was it determined or not? It is confusing.
Author: nd: not determined, fixed in Table 2.
7% rate in 239 lines has been commented by referring to reference number 19 (Because the fat content (7%) of GTP was lower than that of butter [19]).
Part of the results mentioned in section 3.1 are repeated in section 3.2. Authors must revise it and correct it.
Author: Last sentence deleted in Section 3.1 in revised article
Figure 2 in Line 241 must be Figure 3.
Author: Ok. Thank you. It was corrected in the revised text.
From the results, I cannot understand the reason for the higher pH for the butter with a 2% of GTP, and not for 3%. Neither, why did it decrease with the storage period? Can the authors explain it?
Author: As the GTP ratio increased in the butter samples, the pH increased (first day revised Table 2, and Figure 3). During storage, the pH may have decreased since the acidity values increased (data is not given in the text). The activities of the starter culture may have caused this result. This information has been added to the revised text.
What is the explanation for this paragraph While the amount of β-carotene was higher in cow milk-derived butter, it was not found in sheep and goat milk butter. There is no relationship with the content of the article, because only butter derived of cow milk was used.
Author: It was corrected in the revised manuscript.
Authors must revise these two paragraphs for understanding the results: In general, an increase in L* and a* values and b* values did not change. During storage, the samples were found to be lower than the values at the beginning of storage. An increase means a change, and the samples cannot be lower than values.
Author: This section was corrected and some sentences were deleted in the revised text.
In Table 3, for which butter are the values for colour analysis during the storage days?
Author: in all butter samples.
Antioxidant capacities must be given in the corresponding units of concentration depending of the method, but not in absorbance. In this way, it is impossible to compare with other values published for other authors.
Author: In this study, we gave the scavenging activities as IC50 values. However, since IC50 values could not be calculated in reducing abilities methods, we had to give it as absorbance value, as in our previous manuscripts. For all reduction abilities or capacity studies, the results are given as absorbance.
If score for the butter with 45 days of storage is not significant with respect to that for 30 days, why is better to select this last?
Author: When other sensory properties are taken into account, 30 days of storage seems more appropriate.
What do the authors think about the green colour of the butters? Although the score is not bad, could it be a limitation for consumption?
Author: Adding 1% GTP to butter did not give a negative flavor. Color and flavor were appreciated compared to other concentrations. It will be interesting for children especially at breakfast.
Some of these products with tea extracts are already in the market, which are the following step in this research?
Author: In this research, dried green tea leaves were used as an antioxidant in butter by turning them into powder form. In addition, it is aimed to be consumed more at breakfast by creating a greenish color, attracting the attention of children. Thus, it was aimed to benefit from the advantages of using 2 important products (butter and green tea) together, and these goals were realized. Although green tea can be used in different forms, GTP has not been used for this purpose before.
23 of the total references are older than 2013. Are all important for the manuscript?
Author: Usually older references relate to the method. Reference number 20 has been removed from the introduction. Required ordering in the text and in the references list has been adjusted.
Comments on the Quality of English Language
In my opinion, it is necessary a revision of language. Some paragraphs are difficult to understand.
Author: Language of the article has been revised and made more understandable.

Reviewer 2 Report
Abstract
Line 18: The term ppm should not be used for concentration units, use mg/kg (100 ppm = 100 mg/kg) or use percentage (100 ppm = 0.01%).
Material and methods
Lines 83-85: Obtaining GTP is not described in sufficient detail. At what temperature/time were the tea leaves dried? How were the dried leaves ground to obtain the powder?
Lines 120-121: Stock solution of 1 mg/mL of what?
Lines 122-123: Something is wrong (repeated) in this sentence “and after completing the volume to 1 mL with distilled water, the volume was made up to 1 mL with distilled water”
Line 124: put the name of chemicals such as potassium ferricyanide K3Fe(CN)6, the same for FeCl3, CuCl2, CH3COONH4, etc. (revise all manuscript)
Line 200: It sounds strange a 9-point hedonic scale with 10 = dislike extremely and 9 = like extremely
Results and discussion
The layout, content and footnotes in Table 2 should be revised. Perhaps, in Table 2, the first two columns should be the raw materials (cream, GTP) and the following should be the butter treatments, to follow a logical order. The inclusion of (n=2)* at the end of Table 2 title is confusing. If what is meant is that the mean and standard deviation values represent 2 replicates it should be stated differently or in another place. Additionally, there is no footnote for *. What do you mean by Butter samples ** (footnote first day). What do you mean by β-carotene*** (footnote average values). Are the other parameters also averages? The indication of different letters for statistical differences is not well explained. Do you mean different letters within the same line or in the same column?
Line 230-231: Generally, we do not say statistically insignificant but rather that there were no statistically significant differences.
The explanations under subheading 3.2. are confusing. Line 242 what is PC? Please be consistent with the use of abbreviations.
English should be revised throughout manuscript.
Concentrations are sometimes expressed in the form mg g-1 and others as mg/g. Please unify.
The use of statistical significance sometimes lacks logic. For example, in the sentence ‘On the other hand, the storage time variable was found to have a significant effect on colour (P < 0.05) and odour (P > 0.05) on sensory properties’. In general, significant is p<0.05 and not significant is p>0.05.
English should be revised throughout manuscript.
Author Response
Reviewer 2 (Answers of Authors)
Comments and Suggestions for Authors
Author: Responses to the Second Referee's comments are in blue (in the revised manuscript and letter).
Abstract
Line 18: The term ppm should not be used for concentration units, use mg/kg (100 ppm = 100 mg/kg) or use percentage (100 ppm = 0.01%).
Author: It was changed
Material and methods
Lines 83-85: Obtaining GTP is not described in sufficient detail. At what temperature/time were the tea leaves dried? How the dried were leaves ground to obtain the powder?
Author: Detailed information was given in the revised manuscript.
Lines 120-121: Stock solution of 1 mg/mL of what?
Author: Before starting the experiments, stock solutions of the samples at 1 mg/mL concentrations are first prepared. Then, these solutions are diluted depending on the situation and used.
Lines 122-123: Something is wrong (repeated) in this sentence “and after completing the volume to 1 mL with distilled water, the volume was made up to 1 mL with distilled water”
Author: The indicated information was rearranged as “For this aim, first, a 1 mg/mL stock solution was prepared. Different concentrations of this stock solution were transferred to glass tubes, and the solution is completed up with 1 ml of distilled water. Then, 2.5 mL of phosphate buffer (0.2 M, pH: 6.6) and 2.5 mL of 1% potassium ferricyanide [K3Fe(CN)6] was added and this mixture was incubated at 50 °C for 20 min.” in the revised manuscript
Line 124: put the name of chemicals such as potassium ferricyanide K3Fe(CN)6, the same for FeCl3, CuCl2, CH3COONH4, etc. (revise all manuscript)
Author: They were made all the manuscript
Line 200: It sounds strange a 9-point hedonic scale with 10 = dislike extremely and 9 = like extremely
Author: Excuse me, it is my bad. Thank you, I changed in the revised text (1)
Results and discussion
The layout, content and footnotes in Table 2 should be revised. Perhaps, in Table 2, the first two columns should be the raw materials (cream, GTP) and the following should be the butter treatments, to follow a logical order. The inclusion of (n=2)* at the end of Table 2 title is confusing. If what is meant is that the mean and standard deviation values represent 2 replicates it should be stated differently or in another place. Additionally, there is no footnote for *. What do you mean by Butter samples ** (footnote first day). What do you mean by β-carotene*** (footnote average values). Are the other parameters also averages? The indication of different letters for statistical differences is not well explained. Do you mean different letters within the same line or in the same column?
Author: The explanations in Table 2 have been corrected in the Revise article. In Table 2, only the analysis results for the parameters performed on the first day are presented. These values are the average of 2 replications (in each replication, 2 parallel studies were studied, in short, they are the average of 4 numbers. see: 2.2. Experimental design). Different letters for statistical differences were corrected in the revised Table 2.
In Table 2, the raw materials (cream, GTP) are placed in the first 2 columns.
Line 230-231: Generally, we do not say statistically insignificant but rather that there were no statistically significant differences.
Author: Thank you. I changed.
The explanations under subheading 3.2. are confusing. Line 242 what is PC? Please be consistent with the use of abbreviations.
Author: Excuse me, it is my bad. Thank you, I changed in the revised manuscript (BHT)
English should be revised throughout manuscript.
Author: Language of the article has been revised and made more understandable
Concentrations are sometimes expressed in the form mg g-1 and others as mg/g. Please unify.
Author: They were changed as mg/g in the revised manuscript
The use of statistical significance sometimes lacks logic. For example, in the sentence ‘On the other hand, the storage time variable was found to have a significant effect on colour (P < 0.05) and odour (P > 0.05) on sensory properties’. In general, significant is p<0.05 and not significant is p>0.05.
Author: It was corrected in the revised manuscript.
Comments on the Quality of English Language
English should be revised throughout manuscript.
Author: Language of the article has been revised and made more understandable

Reviewer 3 Report
Comments
The authors could present the results of in depth qualitative and quantitative phytochemical analysis (for example LC-MS or LC-UV) of the used GTP. Is this a standardised extract? Moreover, an in-depth discussion, focusing on the individual compounds (EGCG and others) will improve the manuscript.
Author Response
Reviewer 3 (Answers of Authors) (green color)
Comments and Suggestions for Authors
Comments
The authors could present the results of in depth qualitative and quantitative phytochemical analysis (for example LC-MS or LC-UV) of the used GTP. Is this a standardised extract? Moreover, an in-depth discussion, focusing on the individual compounds (EGCG and others) will improve the manuscript.
Author: The referee is right. In this study, the specified analyzes were not performed. However, in a report of congee in yogurt, the components of GTP were investigated and as the incorporation rate increased, EGCG and others increased relative to the control. (See: Çakmakçı, S., Polat, A., Ilgaz, Åž., 2019. Catechin contents of probiotic yogurts produced with green tea powder. 3rd International Conference on Advanced Engineering Technologies, Bayburt, Turkey, 19 - 21 September 2019. Proceeding Book, p 294-300, Bayburt University Publications, and Issue No: 25, ISBN: 978-605-9945-24-0
https://openaccess.bayburt.edu.tr/xmlui/handle/20.500.12403/2159

Round 2
Reviewer 1 Report
In my opinion, the manuscript has been improved and it can be published in the journal.
Sincerely
Author Response
Dear Revewer,
Thank you very much for your valuable comments. The final version of the Manuscript is attached. Best regards,

Reviewer 2 Report
In general, the authors have responded adequately to some of the issues raised and have modified and improved the article, but there are still some aspects that need to be corrected.
In table 2, all figures should have the same number of significant digits (usually 2 to the right of the decimal point) (some SDs have 3).
Lines 291, 378: Generally, we do not say statistically insignificant but rather that there were no statistically significant differences.
Lines 164, 177, 189, 200, 331, 332: Some concentrations are still expressed in the form μg mL-1 while most were changed to μg/mL (or mg/kg or mg/mL). Please unify.
Line 374-376: Again, the use of statistical significance sometimes lacks logic in the same sentence and in Table 4. For example, in the sentence ‘On the other hand, the storage time variable was found to have a significant effect on colour (P < 0.05) and odour (P > 0.05) on sensory properties…’. Why significance for colour is p < 0.05 (that means significant differences were found) and in the same sentence significance for odour is p > 0.05 (that means not significant differences were found). When there are statistically significant differences, p<0.05 is entered, while when there are no statistically significant differences, p>0.05 is entered. In addition, according to footnote to Table 4, the level of significance for sensory evaluation is p < 0.01, then why do they indicate p<0.05/p>0.05 in the text?
Minor editing of English language required
Author Response
Reviewer 2 (Answers of Authors) R2
Comments and Suggestions for Authors
In general, the authors have responded adequately to some of the issues raised and have modified and improved the article, but there are still some aspects that need to be corrected.
Author: Thank you very much for your valuable comments.
In table 2, all figures should have the same number of significant digits (usually 2 to the right of the decimal point) (some SDs have 3).
Author: Ok, corrected in the revised Table 2.
Lines 291, 378: Generally, we do not say statistically insignificant but rather that there were no statistically significant differences.
Author: Changed in the revised manuscript. (298, 402-403)
Lines 164, 177, 189, 200, 331, 332: Some concentrations are still expressed in the form μg mL-1 while most were changed to μg/mL (or mg/kg or mg/mL). Please unify.
Author: Changed (lines 167, 180, 192, 203, 343, 344)
Line 374-376: Again, the use of statistical significance sometimes lacks logic in the same sentence and in Table 4. For example, in the sentence ‘On the other hand, the storage time variable was found to have a significant effect on colour (P < 0.05) and odour (P > 0.05) on sensory properties…’. Why significance for colour is p < 0.05 (that means significant differences were found) and in the same sentence significance for odour is p > 0.05 (that means not significant differences were found). When there are statistically significant differences, p<0.05 is entered, while when there are no statistically significant differences, p>0.05 is entered. In addition, according to footnote to Table 4, the level of significance for sensory evaluation is p < 0.01, then why do they indicate p<0.05/p>0.05 in the text?
Author: The sensory analysis results section has been revised and the relevant section has been changed/corrected in the revised manuscript.
Comments on the Quality of English Language
Minor editing of English language required
Author: English checked.
